# Recent Developments in the Inhibition of Bacterial Adhesion as Promising Anti-Virulence Strategy

**DOI:** 10.3390/ijms24054872

**Published:** 2023-03-02

**Authors:** Camilla Pecoraro, Daniela Carbone, Barbara Parrino, Stella Cascioferro, Patrizia Diana

**Affiliations:** Dipartimento di Scienze e Tecnologie Biologiche Chimiche e Farmaceutiche (STEBICEF), Università degli Studi di Palermo, Via Archirafi 32, 90123 Palermo, Italy

**Keywords:** antibiotic resistance, anti-virulence agents, bacterial adhesion, biofilm formation

## Abstract

Infectious diseases caused by antimicrobial-resistant strains have become a serious threat to global health, with a high social and economic impact. Multi-resistant bacteria exhibit various mechanisms at both the cellular and microbial community levels. Among the different strategies proposed to fight antibiotic resistance, we reckon that the inhibition of bacterial adhesion to host surfaces represents one of the most valid approaches, since it hampers bacterial virulence without affecting cell viability. Many different structures and biomolecules involved in the adhesion of Gram-positive and Gram-negative pathogens can be considered valuable targets for the development of promising tools to enrich our arsenal against pathogens.

## 1. Introduction

The development of antibiotic resistance (AMR) in bacteria has constantly increased in the past decade, undergoing a sudden acceleration in recent years due to the COVID-19 pandemic, for which, especially during the first year, there was excessive and incorrect use of antibiotics [1]. Infections caused by bacteria exhibiting AMR are important causes of prolonged hospitalization times, imposing a significant economic burden on national healthcare systems [2]. It was estimated that the global cost of AMR will reach USD 100 trillion by 2050 [3].

Multi-resistant strains belonging to species *Enterococcus faecium*, *Staphylococcus aureus*, *Klebsiella pneumoniae*, *Acinetobacter baumannii*, *Pseudomonas aeruginosa*, and *Enterobacter* spp., acronymically known as the “ESKAPE” pathogens, are representative examples of pathogens responsible for serious and difficult-to-treat nosocomial infections due to their antibiotic resistance [4]. Common antibiotics, affecting bacterial life or growth processes, impose a high selective pressure for the development of multi-drug resistance (MDR); additionally, they often show low selectivity, targeting biochemical and physiological functions in both pathogenic and commensal bacteria. Chronic infections caused by antibiotic-resistant strains are currently one of the leading causes of death. UK health officials recently defined AMR as a “hidden pandemic”, which, if underestimated, could emerge in the wake of COVID-19, resulting in increased morbidity, mortality, and healthcare expenses associated with the emergence of multi-drug-resistant organisms. Since the danger of returning to a pre-antibiotic era is increasingly realistic, the request for new effective therapeutic strategies against resistant strains becomes very urgent [5,6].

A valuable approach to counteract antibiotic resistance consists in targeting bacterial virulence factors, rather than killing pathogens. Virulence factors are bacterial molecules used by pathogens to colonize the host at the cellular level. These factors can be secretory, cytosolic, or membrane-associated. Secreted factors are important tools used by bacteria to escape the host’s innate and adaptive immune response; cytosolic factors are involved in metabolic, physiological, and morphological adaptive processes, whereas membrane-associated virulence factors confer to the bacterium the ability to adhere to host tissues and biomaterials [7]. These last factors, known as adhesins, play critical roles during infection, being fundamental for the early step of adhesion, which is considered the first stage of bacterial pathogenesis as well as biofilm formation [8].

Although anti-adhesins could put selective pressure on some bacterial processes, since adhesion is not required for microbial survival, its inhibition can be considered a promising approach to treat pathogens, probably limiting the onset of drug-resistant strains [9]. Even if it has been hypothesized that anti-adhesion agents will cause much weaker selection for resistance than traditional antibiotics, some studies have demonstrated that bacteria can mutate and develop resistance to anti-adhesion agents [10]. However, the existence of mechanisms of resistance does not necessarily mean that they will spread, becoming a clinical problem. Adhesins, such as many other virulence factors, are often limited to closely related pathogens, and this entails that anti-adhesion agents usually show a narrow spectrum. For this reason, a combination of several agents may be needed to obtain an efficacious treatment against persistent pathogenic bacteria, but, on the other hand, it was observed that the narrow spectrum of most approaches compromises horizontal gene transfer, which is considered a major process for the transmission of resistance.

## 2. Adhesion Proteins as Promising Targets for Preventing Antibiotic Resistance

Both Gram-positive and Gram-negative pathogens have a plethora of proteins and protein assemblies anchored to their cell walls involved in the adhesion to host tissues.

An important feature common to most bacterial adhesins is their high selectivity for target molecules on the host cell surface, which resembles the lock–key model. Only in a few cases was a different behavior observed, such as for example the *Yersinia* adhesin YadA, which is able to bind a variety of host molecules such as collagen, fibronectin, laminin, β1 integrins, and complement regulators [11].

Most bacterial adhesins in both Gram-negative and Gram-positive bacteria are organized as thin thread-like organelles called fimbriae or pili, which are involved in many important bacterial processes, including conjugation, adherence, twitching motility, biofilm formation, and immunomodulation [12]. The structure of Gram-negative pili is well known and consists of noncovalent polymerization of various pilin subunits, for which chaperones and usher proteins are often required. In the less-studied pilus systems of Gram-positive pathogens, heterotrimeric or -dimeric pili are covalently assembled by the transpeptidase enzyme sortase C [13]. Due to their key role in adhesion and their exposure on the cell surface, these structures can be considered promising targets to develop alternative approaches for preventing and treating bacterial infections; additionally, the immunogenic properties of pilins make them attractive vaccine candidates [14].

Recently, in addition to pili inhibition, for Gram-negative bacteria, the hampering of the highly oxidizing enzyme DsbA was identified as an attractive strategy to attenuate the virulence of relevant pathogens, such as *E. coli* and *Salmonella enterica*. In Gram-negative pathogens, in fact, the disulfide bond (Dsb) system assumes a key role in bacterial pathogenesis by catalyzing disulfide bond formation in the production of several bacterial proteins, including adhesins, flagellae, toxins, and other virulence factors [15].

Despite this, the involvement of DsbA in several virulence factors of relevant pathogens is a cause of concern since it can generate high selective pressure on bacterial life processes; these enzymes involved in the biogenesis of adhesins currently represent interesting target candidates to obtain anti-virulence agents with anti-adhesion mechanisms. It was observed that the deletion of *dsbA*/*dsbB* genes in numerous Gram-negative pathogens results in a significant reduction in virulence, and DsbA inhibition in Gram-negative bacteria causes a drastic decrease in virulence and increased sensitivity to antibiotics. Multiple different DsbA homologs have been described, and among them, the DsbA oxidative system found in *E. coli* (EcDsbA) is the best known. Two enzymes are involved in this system: (i) DsbA, which oxidizes unfolded polypeptides via a disulfide exchange reaction; and (ii) DsbB, which restores the oxidizing activity of DsbA by regenerating a disulfide group at the DsbB active site [16]. EcDsbA is a thioredoxin (TRX)-like thiol oxidase whose redox-active site is formed by the residues Cys30-Pro31-His32-Cys33, which is close to a hydrophobic groove required for binding the cognate oxidase EcDsbB.

Whereas in Gram-negative bacteria pili play a major role in adhesion, in Gram-positive pathogens, this function is performed by a class of proteins with structural motifs similar to those of pilin components known as microbial surface components recognizing adhesive matrix molecules (MSCRAMMs). These surface proteins, covalently linked to the peptidoglycan by the transpeptidase sortase A (SrtA), include important host extracellular proteins such as protein A (Spa), fibronectin-binding proteins (FnbpA, FnbpB), clumping factors (ClfA, ClfB), a collagen-binding protein (Cna), and three serine-aspartate repeat proteins (SdrC, SdrD, SdrE). Representative examples of MSCRAMMS involved in the adhesion process are reported in Table 1.

Undoubtedly, among the different MSCRAMMs, Fnbps have been the most investigated and were found to be crucial especially in the staphylococcal adhesion process. All MSCRAMMs share a common structural motif known as the LPXTG motif, which was recognized by SrtA. This recognition marks the starting point for two sequential reactions, a thioesterification and a transpeptidation, that lead to the formation of an amide linkage of the C-terminal Thr of the protein to pentaglycine cross-bridges in *S. aureus*.

Since MSCRAMMs are involved in many important bacterial functions and pathogenic processes, including adhesion, colonization, and evasion of innate immune response, SrtA has attracted a growing interest in the medicinal field as an ideal target for the development of effective anti-virulence agents [17]. The inhibition of this enzyme allows obtaining efficacious anti-adhesion agents with anti-virulence profiles, since SrtA is not indispensable for microbial growth and viability. Additionally, with SrtA being a membrane enzyme, it can be reached more easily than intracellular bacterial targets, and its inhibition proved to be advantageous also in terms of toxicity, since there are no eukaryotic homologs. 

Since teichoic acids (TA), essential wall constituents of staphylococci, play a key role in the adhesion to the host tissue and in biofilm formation, they can be considered valuable targets to develop an anti-adhesion strategy. It was observed that the interference with WTA biosynthesis led to a strong decrease in the ability of *S. aureus* and *B. subtilis* strains to establish infection in animal models [18]. 

## 3. Recent Developments of Anti-Adhesion Agents against Relevant Pathogens

Since the adhesion of relevant pathogens to mucosal surfaces is a crucial step for the pathogenesis of many infections of the respiratory, urinary, and gastrointestinal tracts, the identification of efficacious anti-adhesion agents has posed a great challenge in the medical field in the past years. Many strategies have been proposed to reduce bacterial adhesion in order to hinder the infection process.

Among the Gram-negative pathogens responsible for serious chronic infections, *E. coli* and *P. aeruginosa* represent a remarkable risk for human and animal health due to their spread to waterways and other environmental sources. Different crucial adhesion mechanisms have been identified for these pathogens in recent years, and most of these are involved in biofilm formation.

Numerous adhesins and extracellular matrix components, including flagella and curli fibers, have a key role in *E. coli* biofilm formation. Flagella are essential for the transport and adhesion of bacteria to a surface, whereas the proteinaceous curli fibers have a double role, as in addition to being required in the initial stages of *E. coli* attachment to the host cells, they are also a major component of the *E. coli* biofilm matrix [19,20].

Type 1 and P pili, assembled by the chaperone–usher pathway (CUP pili), are recognized as important adhesive surface structures fundamental in *E. coli* adhesion and infection [21]. During the pathogenesis of urinary tract infections (UTIs), type 1 pili mediate interactions between uropathogenic *E.coli* (UPEC) with the bladder, whereas P pili target the kidney [22,23]. Pinkner et al. described the design of new pyridone compounds as efficient pilicides able to prevent CUP pilus assembly and to decrease type 1 pilus-dependent biofilm formation. In particular, the pilicide ec240 proved to be able to block S and P pilus assembly and to disrupt the regulatory connections between type 1 pili and flagella [24,25]. In vivo experiments demonstrated that uropathogenic *E. coli* (UPEC) strains lacking type 1 or P pili were markedly attenuated in their ability to cause urinary tract infections [26]. 

Structurally, type 1 pili are formed by assembled FimA subunits, which form a rod structure tipped with FimH adhesin molecules, responsible for a specific adhesion to mannose residues on epithelial cells, thereby facilitating infection. FimH adhesin confers the ability of *E. coli* to autoaggregate, a property that strongly contributes to the colonization of mammalian hosts by pathogenic bacteria [27].

For the secretion of proteins involved in communication, virulence, and adhesion, Gram-negative bacteria have evolved different secretion systems (I-IX). Some of these systems are formed by multiple proteins building a complex spanning the cell envelope, whereas the type V secretion system is rather minimal. Proteins of the type V secretion system are known as autotransporters (ATs) to indicate a self-sufficient system for secretion [28]. In particular, autotransporters (ATs) are outer membranes/secreted proteins structurally characterized by the presence of three distinct domains: (i) an N-terminal passenger domain, which typically mediates the export of the protein across the cytoplasmic membrane; (ii) a surface-localized protein known as the passenger or α-domain; and (iii) a carboxy-terminal domain, known as the β-barrel translocator domain, which facilitates the secretion of the passenger domain through the outer membrane. These properties allow their independent transport across the bacterial membrane system and their final routing to the cell surface [29].

Paxman et al. identified the inhibition of the autotransporter (AT) adhesin UpaB as an effective approach to hamper the adhesion of uropathogenic *E. coli* (UPEC) to extracellular matrix (ECM) proteins and prevent the dangerous colonization of the urinary tract [30]. Another approach involves the use of compounds acting as analogs of bacterial adhesin receptors [31]. A treatment with α-mannoside, able to interfere with type 1 fimbriae FimH1, was suggested for the treatment of catheter-related urinary tract infections caused by UPEC [32]. The mechanism of FimH inhibition is based on the structural similarity between mannose and mannosylated receptors sited in urothelial surfaces. In fact, the binding of d-mannose to bacterial FimH prevents bacterial attachment to mannosylated receptors in the urinary tract. 

Recently, Ortiz and coworkers analyzed the effects of natural and synthetic antimicrobials, including carvacrol, oregano extract, brazilin, palo de Brasil extract, and rifaximin, on the adherence of different enterohemorrhagic (EHEC), enteroaggregative (EAEC), and enteroaggregative hemorrhagic (EAHEC) strains (EHEC O157:H7, EAEC 042, and EAHEC O104:H4) to HEp-2 cells [33]. The treatment of the aggregative strains with antimicrobials at sub-minimum bactericidal concentrations (MBCs), in particular with carvacrol (0.010 mg/mL), caused a significant alteration of the characteristic stacked-brick structure. The change in bacteria–bacteria adhesion observed was due to the modification in gene expression in *E. coli,* consisting in the downregulation of *aggR*, *pic*, and *aap*, and the upregulation of *aggA*.

In *P. aeruginosa,* biofilm formation can be promoted by the adhesive action of several components, such as flagella, type IV pili, Cup fimbriae, extracellular DNA, and Psl polysaccharide. Many of these components are also important elements of the EPS in mature biofilm. The exopolysaccharide cell-surface-associated Psl acts as an adhesin in the initial phase of biofilm formation, but later in the biofilm cycle, it functions as a peripheral exopolysaccharide [34].

Among the different virulence factors involved in the adhesion of Gram-negative bacteria, LecA and LecB have been widely studied as targets to obtain compounds able to interfere with the adhesion of *P. aeruginosa.* Recently, Titz et al. reported the synthesis of a new class of C-glycosidic LecB inhibitors [35]. The most active compounds, **1a,b** (Figure 1), exhibited IC_50_[LecB _PAO1_] values in the range of 1.32–1.87 μM and IC_50_[LecB _PA14_] in the range of 0.20–0.33 μM. The crystallographic structure of the dimethylthiophene **1a** with LecB_PA14_ elucidated its binding mode characterized by lipophilic interactions of the methyl group in the ortho-position to the sulfonamide with hydrophobic protein residues in the binding pocket.

As previously discussed, recently, the highly oxidizing periplasmic enzyme DsbA, acting as a folding catalyst for different virulence factors, was described as crucial for the virulence of relevant Gram-negative pathogens [36]. Small molecules, belonging to different chemical classes, including phenylthiophene and phenoxyphenyl, have recently been reported by Totsika et al. as DsbA inhibitors able to interfere with UPEC and *S. enterica* motility [14].

Many efforts have been made in recent years in order to develop new compounds capable of eradicating serious staphylococcal infections by inhibiting the biofilm formation process interfering with bacterial adhesion. *S. aureus* is one of the main causes of persistent human infections, which, in 2017, was categorized as a high-priority multi-drug-resistant (MDR) pathogen by the World Health Organization (WHO). Biofilm is currently considered one of the most relevant bacterial virulence factors, which significantly contributes to microbial survival in hostile environments, and it is one of the major causes of *S. aureus* antibiotic resistance and pervasiveness [37,38]. Many series of indole compounds, belonging to different chemical classes, including nortopsentin analogs [39], topsentin analogs [40], and imidazothiadiazole derivatives [41], were recently reported as potent inhibitors of *S. aureus* biofilm formation. The most potent compounds of each series showed biofilm inhibitory concentration (BIC_50_) values lower than 1 µM at least against one staphylococcal strain tested. In particular, thiazole nortopsentin analogs **2a** and **2b** (Figure 2) elicited BIC_50_ values against *S. aureus* ATCC25923 of 0.4 and 0.5 µM, respectively, whereas the 1,2,4-oxadiazole topsentin derivative **3** (Figure 2) displayed a BIC_50_ value of 0.2 µM towards the same bacterial strain. In the imidazothiadiazole series, compounds **4a–c** (Figure 2) potently inhibited biofilm formation in *S. aureus* ATCC25923 and ATCC6538, with BIC_50_ values ranging from 0.5 to 1.8 µM. Interestingly, for the oxadiazole topsentin class, the mechanism of biofilm inhibition was identified, and it consists in a strong SrtA inhibition (IC_50_ values in the range of 2.2–10 µM).

SrtA has attracted great attention in recent years in order to develop new anti-virulence compounds, mainly with anti-biofilm properties [42]. Many efforts have been made with the aim to identify new sortase A inhibitors through the screening of natural products or small compound libraries [43]. Recently, 27 derivatives were identified as covalent inhibitors of *S. aureus* SrtA by screening a library of 50,000 compounds using a Förster resonance energy transfer (FRET) assay followed by NMR-based validation and binding reversibility analysis [44]. The new inhibitors were classified into seven classes, chemically different, of which five were previously reported as nitriles, pyridazinones, thioamides, Michael acceptors, and aryl (β-amino)ethyl ketones, and two new classes were characterized by the presence of N-hydroxy/N-amino sulfonamide group and an activated halogen group, respectively. Except for pyridazinones, which bound the active site of the enzyme reversibly, the other inhibitor covalently bound the Cys184 residue.

## 4. Conclusions

Chronic infections often due to biofilm-forming pathogens remain a major healthcare concern, associated with a high social and economic burden. There is an urgent need for new effective therapeutic strategies able to overcome the major MDR mechanisms in order to avoid the possibility of returning to a pre-antibiotic era. Among the different strategies proposed in recent years, the anti-virulence approach may be considered one of the most promising. It consists in depriving the bacterium of its virulence without interfering with its viability; this allows imposing a low selective pressure for the onset of new MDR mechanisms. Bacteria exhibit a plethora of virulence factors, many of which are involved in adhesion. Counteracting bacterial adhesion to surfaces blocks both the pathogenesis and the ability of the bacterium to form biofilm, which strongly contributes to making bacteria resistant to antibiotics. In order to develop an effective anti-adhesion strategy, it is necessary to fully understand the mechanisms of pathogen–host interaction, which is a very complex and dynamic process. To obtain a successful therapy, it may be useful to administer combinations of different pathoblockers, capable of acting against different virulence factors, or combinations of these with traditional antibiotics to reduce the bacterial load in high-risk patients.

About the possible strategies proposed for the treatment of AMR infections caused by Gram-positive bacteria, SrtA inhibition can be considered one of the most valid approaches, even if the real efficacy of this strategy needs confirmation in in vivo models of infections. The inhibition of this transpeptidase has another benefit: hitting a single target results in the inhibition of numerous proteins involved in adhesion (MSCRAMMs). 

Concerning the adhesion mechanisms of Gram-negative pathogens, many targets can be evaluated to develop effective therapies; among them, a not fully developed research area but with great potential could be the recognition of molecules capable to inhibit the DSB oxidative protein folding machinery, which could lead to a promising anti-virulence strategy for disarming the bacteria rendering them unable to cause infection. Since the crystallographic structures of the two enzymes, SrtA and DsbA are known, it would be interesting to design, through computational studies, molecules bearing the same structure and chemical groups capable of binding both active sites. This would allow obtaining derivatives able to interfere at the same time with the adhesion of relevant Gram-positive and Gram-negative pathogens.

Anti-adhesion agents may find different therapeutic applications, both alone in the prophylaxis of medical surgery or for the coatings of medical devices and as antibiotic adjuvants. The discovery of new molecules showing activity towards bacterial non-essential targets represents a valuable approach to developing antibiotic adjuvants that can be used in combination with antibiotics to minimize the impact of antibiotic resistance [45].

## Figures and Tables

**Figure 1 ijms-24-04872-f001:**
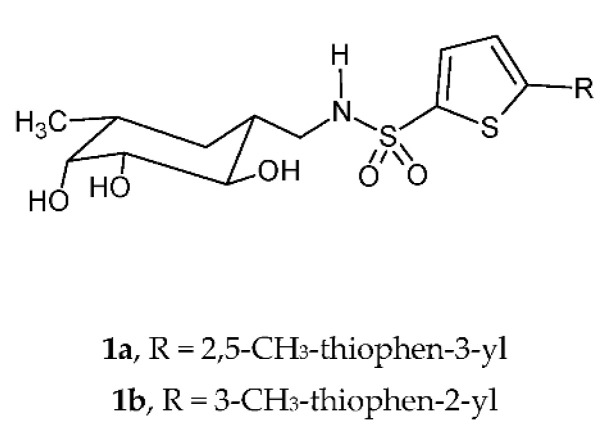
Chemical structures of C-glycosidic LecB inhibitors **1a,b**.

**Figure 2 ijms-24-04872-f002:**
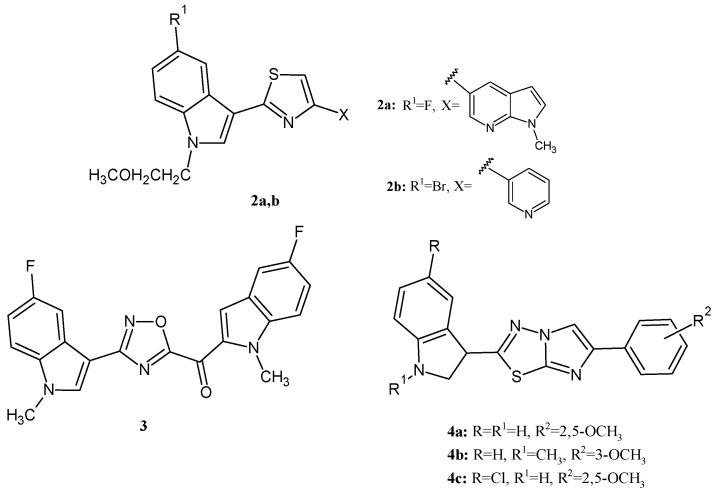
Indole compounds **2–4** with potent inhibitory activity against staphylococcal biofilm formation.

**Table 1 ijms-24-04872-t001:** Examples of the main MSCRAMMs with adhesive properties.

Proteins	Functions	Role in Pathogenesis
Protein A (Spa)	Binds Fc domain for immunoglobulins; binds complement protein C3	Inhibition of innate and adaptive immune responses
Fibronectin-binding protein A (FnbpA)	Adhesin for fibrinogen, fibronectin and elastin	Adhesion; colonization; biofilm formation
Fibronectin-binding protein homolog (FnbpB)	Adhesin for fibronectin and elastin	Adhesion; colonization; biofilm formation
Clumping factor A (ClfA)	Platelet adhesion (fibrin-mediated)	Adhesion; colonization; evasion of innate immune defenses
Clumping factor B (ClfB)	Platelet adhesion (fibrin-mediated)	Adhesion; colonization; evasion of innate immune defenses
Collagen-binding protein (Cna)	Adhesin for collagen (type I and IV)	Adhesion
Serine-aspartate repeat protein C (SdrC)	Adhesin	Adhesion; colonization
Serine-aspartate repeat protein D (SdrD)	Adhesin	Adhesion; colonization
Serine-aspartate repeat protein E (SdrE)	Adhesin	Adhesion; colonization

## Data Availability

Not applicable.

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
