# Peer review of "Recent Developments in the Inhibition of Bacterial Adhesion as Promising Anti-Virulence Strategy"

_ijms, 2023, doi:10.3390/ijms24054872_

Round 1

Reviewer 1 Report

Pecoraro et al. present an opinion piece on antivirulence strategies targeting bacterial adhesion. This is an important topic that has gained increasing attention in the past years. While the topic is timely, I have some reservations about the paper as it stands, with three major points of criticism:

1. The manuscript is offered as an opinion piece. However, I fail to see a clear opinion expressed. It does not promote a particular viewpoint or add any original ideas to the existing literature. It might be better described a mini-review or similar. 

2. While the paper covers advances made in anti-adhesion work, I was rather dismayed to see that pilicides (see doi: 10.1073/pnas.0606795103, 10.1128/mBio.02038-14) were not included in the discussion at all. This is a major omission, as this class of molecule is among the most promising for antivirulence therapy.

3. Regarding Gram-negative bacteria, only fimbriae and pili are discussed in any detail in section 2. However, non-fimbrial adhesins are alluded to later. Non-fimbrial adhesins, such as autotransporters or small beta-barrel proteins, should be discussed here.

Minor comments:

1.  Scientific names are not always given in italics (e.g. lines 73, 100, 195, 203-209).

2. Some abbreviations (BIC, FRET, MBC, EHEC, EAEC) are not defined in the text.

3. Line 22: a reference for this statement is needed.

4. line 27: 'strains' does not seem correct here as the list is of species. However, the only certain strains within these species are MDR. I suggest changing to, "Multi-resistant strains belonging to the species Enterococcus..."

5. line 33: often show low selectivity

6. line 44: 'Secretory' refers to the components of the cellular machineries needed for secretion (secretory pathway); I would change this to 'secreted'.

7. Lines 53-54: This is speculative, and it cannot be ruled out that anti-adhesive therapies would not have an effect on commensals. The may be unlikely to have an effect, but experimental verification would be neede.

8. Line 59: Though this is correct for many adhesins, there are also examples of promiscuous adhesins that bind to a variety of surfaces/receptors (e.g. the Yersinia adhesin YadA). This should be acknowledged.

9. Line 64: consists of

10. Line 78: this function is performed

11. Table 1: FnpbA stands for fibronectin-binding protein A, surely. And note spelling of collagen.

12. Line 95: reached more easily than intracellular targets

13. Line 116: being required in the initial stages

14: Line 130: autotransporters (and other non-fimbrial adhesins) should be introduced earlier.

15: Line 130: here the UpaB protein is being referred to, so the italics are not needed.

16: Line 149: Cup fimbriae

17: Line 152: initial phase of biofilm formation, but later in the biofilm cycle functioning as a peripheral exopolysaccharide.

18. Lines 155-156: There is some controversy regarding MAM7 as a cell-surface exposed adhesin, as its orthologue YebF is a periplasmic protein involved in lipid trafficking from the inner membrane to the outer membrane. (see 10.1038/s41598-017-09111-6). Also, this seems out of place and no anti-virulence strategies/compounds relating to MAMs are mentioned.

19. Lines 172-189: This seems out of place and much of it repeats material from section 2. 

20: line 176: gene names in italics

21. Line 208: Fig. 2?

22: lines 228-231: This seems out of place here.

Author Response

Comments and Suggestions for Authors

Pecoraro et al. present an opinion piece on antivirulence strategies targeting bacterial adhesion. This is an important topic that has gained increasing attention in the past years. While the topic is timely, I have some reservations about the paper as it stands, with three major points of criticism:

  1. The manuscript is offered as an opinion piece. However, I fail to see a clear opinion expressed. It does not promote a particular viewpoint or add any original ideas to the existing literature. It might be better described a mini-review or similar. 

We agree with the reviewer's observation. We prepared this manuscript as invited Mini-review for the special issue "Enzymes and Enzyme inhibitors-Applications in Medicine and Diagnosis 2.0". Unfortunately, despite the guest Editor had agreed to this type of contribution, the journal does not provide this type of article, therefore we had to choose an article type that could fit. Despite that, in the revised version we clarified our point of view highlighting, especially in the conclusions, the anti-adhesion strategies that we reckon more promising. 

  1. While the paper covers advances made in anti-adhesion work, I was rather dismayed to see that pilicides (see doi: 10.1073/pnas.0606795103, 10.1128/mBio.02038-14) were not included in the discussion at all. This is a major omission, as this class of molecule is among the most promising for antivirulence therapy.

We added the two reference suggested Pinkner, J.S.; Remaut, H.; Buelens, F.; Miller, E.; Åberg, V.; Pemberton, N.; Hedenström, M.; Larsson, A.; Seed, P.; Waksman, G.; et al. Rationally Designed Small Compounds Inhibit Pilus Biogenesis in Uropathogenic Bacte-ria. Proceedings of the National Academy of Sciences 2006, 103, 17897–17902 and  Greene, S.E.; Pinkner, J.S.; Chorell, E.; Dodson, K.W.; Shaffer, C.L.; Conover, M.S.; Livny, J.; Hadjifrangiskou, M.; Almqvist, F.; Hultgren, S.J. Pilicide Ec240 Disrupts Virulence Circuits in Uropathogenic Escherichia Coli. mBio 2014, 5, e02038-14, in the section 3 of the manuscript.

  1. Regarding Gram-negative bacteria, only fimbriae and pili are discussed in any detail in section 2. However, non-fimbrial adhesins are alluded to later. Non-fimbrial adhesins, such as autotransporters or small beta-barrel proteins, should be discussed here.

We added a brief discussion and two new references to introduce the non-fimbial adhesins (lines 139-149) as required.

Minor comments:

  1. Scientific names are not always given in italics (e.g. lines 73, 100, 195, 203-209).

Thanks for the suggestions, we checked the text and we corrected names as required.

  1. Some abbreviations (BIC, FRET, MBC, EHEC, EAEC) are not defined in the text.

We defined the abbreviations in their first appearance in the text as required.

  1. Line 22: a reference for this statement is needed.

We agreed with this comment and we added the recent reference on the topic: Christaki, E.; Marcou, M.; Tofarides, A. Antimicrobial Resistance in Bacteria: Mechanisms, Evolution, and Persistence. J Mol Evol 2020, 88, 26–40, doi:10.1007/s00239-019-09914-3.

  1. line 27: 'strains' does not seem correct here as the list is of species. However, the only certain strains within these species are MDR. I suggest changing to, "Multi-resistant strains belonging to the species Enterococcus..."

We modified the sentence as suggested

  1. line 33: often show low selectivity

We corrected “showed” in “show” as required

  1. line 44: 'Secretory' refers to the components of the cellular machineries needed for secretion (secretory pathway); I would change this to 'secreted'.

Thanks for the suggestion, we changes the sentence as suggested

  1. Lines 53-54: This is speculative, and it cannot be ruled out that anti-adhesive therapies would not have an effect on commensals. The may be unlikely to have an effect, but experimental verification would be neede.

We modified the sentence as follow: “Furthermore, anti-adhesion strategy is advantageous also in terms of side effects, as an ef-fect on commensal microbes is unlikely.”

  1. Line 59: Though this is correct for many adhesins, there are also examples of promiscuous adhesins that bind to a variety of surfaces/receptors (e.g. the Yersinia adhesin YadA). This should be acknowledged.

We accepted the reviewer's advice and amended the paragraph as follows: “An important feature, common to most of bacterial adhesins, is their high selectivity for target molecules on host cell surface, which reminds the lock-key model.[9] Only in few cases a different behavior was observed, such as for example the Yersinia adhesin YadA, which is able to bind a variety of host molecules such as collagen, fibronectin, laminin, β1 integrins, and complement regulators”. A reference on the topic was also added.

  1. Line 64: consists of

The error has been corrected

  1. Line 78: this function is performed

We modified the sentence as required

  1. Table 1: FnpbA stands for fibronectin-binding protein A, surely. And note spelling of collagen.

We corrected the mistakes in Table 1 as suggested

  1. Line 95: reached more easily than intracellular targets

The sentence was modified

  1. Line 116: being required in the initial stages

We changed requested in required as suggested

14: Line 130: autotransporters (and other non-fimbrial adhesins) should be introduced earlier.

We added a new paragraph and two new references to introduce the information required

15: Line 130: here the UpaB protein is being referred to, so the italics are not needed.

We modified the style

16: Line 149: Cup fimbriae

We fixed the typo

17: Line 152: initial phase of biofilm formation, but later in the biofilm cycle functioning as a peripheral exopolysaccharide.

We modified the sentence as suggested

  1. Lines 155-156: There is some controversy regarding MAM7 as a cell-surface exposed adhesin, as its orthologue YebF is a periplasmic protein involved in lipid trafficking from the inner membrane to the outer membrane. (see 10.1038/s41598-017-09111-6). Also, this seems out of place and no anti-virulence strategies/compounds relating to MAMs are mentioned.

Thank you for the suggestion, we share this opinion therefore we decided to cut the sentence in this revised version.

  1. Lines 172-189: This seems out of place and much of it repeats material from section 2.

We revised the text moving the discussion on the Dsb system in chapter 2 in order to avoid repetitions and fragmentations.

20: line 176: gene names in italics

We corrected the typo

  1. Line 208: Fig. 2?

We made the correction

22: lines 228-231: This seems out of place here.

Thanks for the suggestion. We decided to delete the sentence since no inhibitor of BspC adhesin was discussed in the manuscript.

Reviewer 2 Report

Journal:              IJMS (ISSN 1422-0067)

Manuscript ID:  ijms-2210138

Type:                 Opinion

Title:                 Recent development in the inhibition of bacterial adhesion in the fight against antibiotic-resistance

Camilla Pecoraro1, Daniela Carbone1, Barbara Parrino1, Stella Cascioferro1* and Patrizia Diana1

Comments

The theme of the article is of global importance and is being researchers vigorously.

There are a few issues which need clarification and supporting evidence.

Q1.   Title conveys “ … fight against antibiotic resistance.”

         The proposed antiadhesion strategy and the associated details do not elucidate how it will help to counter antibiotic resistance. Hence, the title needs to be revised to match the contents of the article.

Q2.  Like the emergence of antibiotic Resistance, there is always a possibility of bacteria undergoing mutation and developing resistance to anti-adhesion agents.

Q3.   How many factors regulate the adhesion process?

        What is the frequency with which adhesion negative mutants emerge and undergo reversion.

Q4.   Lines 56-57

Both Gram-positive and Gram-negative pathogens have a plethora of proteins and protein assemblies, anchored to their cell walls, involved in the adhesion to the host tissues.

       How many proteins and protein assemblies do we need to handle to develop a viable anti-adhesion strategy?

Q5.  Lines 60-62

Most bacterial adhesins, in both Gram-negative and Gram-positive bacteria, are organized as thin thread-like organelles called fimbriae or pili, which are involved in many important bacterial processes, including conjugation, adherence, twitching motility, biofilm formation and immunomodulation.[10]

       Anti-adhesin may not affect the growth but is likely to put selective pressure on other bacterial processes.  

Q6. Lines 72-76

DsbA, was identified as an attractive strategy to attenuate the virulence of relevant pathogens, such as E. coli and Salmonella enterica. In Gram-negative pathogens, in fact, DsbA assumes a key role in the bacterial pathogenesis by catalyzing disulfide bond formation in the production of several bacterial proteins, including adhesins, flagellae, toxins, and other virulence factors [13]

       The involvement of DsbA in several bacterial proteins involved in the virulence of relevant pathogens is a cause of concern since it can generate high selective pressure on bacterial life processes.

Q7.  Line 86: Table 1. Examples of the main MSCRAMMs with adhesive properties.

It implies that multiple factors need to be inhibited to achieve an anti-adhesion target.

If the target is the genes inhibition or intracellular molecules then it is difficult to stop the production of adhesive molecules.

The target should be the molecules released in the vicinity of the pathogenic bacteria.

Q8. Lines 236-237: Among the different strategies proposed in the last years, the antivirulence approach may be considered one of the most promising.

       This proposal/opinion has been made by various researchers. In the last three decades, quite a few Review and Opinion articles have focused on it as one of the most promising candidates for developing “antivirulence” strategies.

       So, it is recommended that this article may not be categorized as an Opinion article. Instead, it should focus on providing an update on the Anti-adhesion research status. And suggest whether it can hold well as a standalone strategy or need complementation with other processes.

Q9.  References

        Only 1 Reference pertains to those published in 2022.

        And only 5 References are from 2021.

       If the interest in this area is declining among scientists, then it reflects that the area is losing its importance.  

Author Response

Comments

The theme of the article is of global importance and is being researchers vigorously.

There are a few issues which need clarification and supporting evidence.

Q1. Title conveys “ …fight against antibiotic resistance.”

The proposed antiadhesion strategy and the associated details do not elucidate how it will help to counter antibiotic resistance. Hence, the title needs to be revised to match the contents of the article.

Thank you for the suggestion. We modified the title in “Recent development in the inhibition of bacterial adhesion as promising anti-virulence strategy”

Q2.  Like the emergence of antibiotic Resistance, there is always a possibility of bacteria undergoing mutation and developing resistance to anti-adhesion agents.

This is a possibility, but these agents not acting by killing or arresting growth of the pathogens, impose a low selective pressure for the development of resistance. We find this observation very interesting, but it would be difficult to go deeper into the subject in this type of manuscript.

Q3. How many factors regulate the adhesion process?

What is the frequency with which adhesion negative mutants emerge and undergo reversion.

We consider the reviewer's observations very interesting and worthy of further study, but the type of manuscript does not allow us to deal in detail with all factors involved in the regulation of adhesion mechanisms. We have dealt with what, in our opinion, can be considered promising targets to obtain anti-adhesion agents. We reckon that this would be an interest topic for a more comprensive review.

Q4. Lines 56-57

Both Gram-positive and Gram-negative pathogens have a plethora of proteins and protein assemblies, anchored to their cell walls, involved in the adhesion to the host tissues.

How many proteins and protein assemblies do we need to handle to develop a viable anti-adhesion strategy?

As we reported in the manuscript, different proteins can be considered target to develop an anti-adhesion strategy. In the case of Gram-positive bacteria, for example, the inhibition of SrtA leads to the inhibition of numerous other proteins (MSCRAMMs) involved in the adhesion. In this way, hampering only one enzyme we can have a significant inhibition of different targets involved in the adhesion. We modified the text in the conclusion to highlight this benefit of SrtA inhibitors in order to give the information required by the reviewer.

  Q5.  Lines 60-62

Most bacterial adhesins, in both Gram-negative and Gram-positive bacteria, are organized as thin thread-like organelles called fimbriae or pili, which are involved in many important bacterial processes, including conjugation, adherence, twitching motility, biofilm formation and immunomodulation.[10]

Anti-adhesin may not affect the growth but is likely to put selective pressure on other bacterial processes.  

We modified the text (lines 54-55) as suggested: “Although anti-adhesins could put selective pressure on bacterial processes, since adhesion is not required for microbial survival, its inhibition can be considered a promising approach to treat multidrug resistant infection…”

 Q6. Lines 72-76

DsbA, was identified as an attractive strategy to attenuate the virulence of relevant pathogens, such as E. coli and Salmonella enterica. In Gram-negative pathogens, in fact, DsbA assumes a key role in the bacterial pathogenesis by catalyzing disulfide bond formation in the production of several bacterial proteins, including adhesins, flagellae, toxins, and other virulence factors [13]

The involvement of DsbA in several bacterial proteins involved in the virulence of relevant pathogens is a cause of concern since it can generate high selective pressure on bacterial life processes.

We focused on the pleiotropic effects of the depletion of the DsbA/B thiol oxidation system on multiple virulence-associated phenotypes and on the decreased capacity of pathogens to establish infections in animal models. We do not find experimental evidences on the possibility to generate high selective pressure on bacterial life processes although we do not exclude this could happen.

Q7.  Line 86: Table 1. Examples of the main MSCRAMMs with adhesive properties.

It implies that multiple factors need to be inhibited to achieve an anti-adhesion target.

If the target is the genes inhibition or intracellular molecules then it is difficult to stop the production of adhesive molecules.

The target should be the molecules released in the vicinity of the pathogenic bacteria.

Thank you for the interesting comment, but in the manuscript we report the SrtA inhibition as strategy to impeed the covalent anchoring of MSCRAMMs on the bacterial cell wall, we do not discuss the gene inhibition. We modified the text in order to better clarify the mechanism of this inhibition.

Q8. Lines 236-237: Among the different strategies proposed in the last years, the antivirulence approach may be considered one of the most promising.

This proposal/opinion has been made by various researchers. In the last three decades, quite a few Review and Opinion articles have focused on it as one of the most promising candidates for developing “antivirulence” strategies.

So, it is recommended that this article may not be categorized as an Opinion article. Instead, it should focus on providing an update on the Anti-adhesion research status. And suggest whether it can hold well as a standalone strategy or need complementation with other processes.

We agree with the reviewer's observation. We prepared this manuscript as invited Mini-review for the special issue "Enzymes and Enzyme inhibitors-Applications in Medicine and Diagnosis 2.0". Unfortunately, despite the guest Editor had agreed to this type of contribution, the journal does not provide this type of article, therefore we had to choose an article type that could fit. For what concern the suggestion on a possible therapeutic application we added a sentence in the conclusion.

Q9. References

 Only 1 Reference pertains to those published in 2022.

And only 5 References are from 2021.

If the interest in this area is declining among scientists, then it reflects that the area is losing its importance.  

We added four new recent references (. Molecular Aspects of Medicine 2021, 81, 100998; Antibiotics 2023, 12, 195; Bioorganic Chemistry 2023, 131, 106307, World J Microbiol Biotechnol 2022, 39, 18)

Round 2

Reviewer 1 Report

The authors have addressed most of my concerns satisfactorily. I only have a few minor points for the authors to consider:

1. Line 66: The reference provided for YadA is out of date and there are more recent reviews on this protein, e.g. 10.1016/j.ijmm.2014.12.008.

2. Line 82: the disulphide bond system.

3. Line 113: I don’t think the LPXTG motif is a structural one.

4. Line 172: the reference provided does not seem to be appropriate for type 5 secretion systems in general. There are several recent reviews on the topic that would be better.

5. Lines 173-176: the explanation here is a bit confusing and should be rephrased.

6. Lines 223-224: The species is in point of fact Salmonella enterica (as noted correctly on line 82). Typhimurium is a serotype of S. enterica, so the correct notation would be S. enterica serotype Typhimurium, or at the very least S. Typhimurium (the serotype is not given in italics).

7. Lines 283-287: The explanation is again a little confusing. Are the authors suggesting that a single molecule inhibiting both be developed, or that both inhibitors should be applied at the same time? Please clarify what is meant here.

Author Response

The authors have addressed most of my concerns satisfactorily. I only have a few minor points for the authors to consider:

  1. Line 66: The reference provided for YadA is out of date and there are more recent reviews on this protein, e.g. 10.1016/j.ijmm.2014.12.008.

Thanks for the suggestion, we replaced the reference with: Mühlenkamp, M.; Oberhettinger, P.; Leo, J.C.; Linke, D.; Schütz, M.S. Yersinia Adhesin A (YadA) – Beauty & Beast. International Journal of Medical Microbiology 2015, 305, 252–258, doi:10.1016/j.ijmm.2014.12.008.

  1. Line 82: the disulphide bond system.

We corrected the typo

  1. Line 113: I don’t think the LPXTG motif is a structural one.

We modified the sentence as follow: “All MSCRAMMs share common structural motifs known as LPXTG motifs (leucine, pro-line, any amino acid, threonine and glycine), which were recognized by Srt A.”

  1. Line 172: the reference provided does not seem to be appropriate for type 5 secretion systems in general. There are several recent reviews on the topic that would be better.

We replaced the old reference with two new ones: 1) Dautin, N. Folding Control in the Path of Type 5 Secretion. Toxins 2021, 13, 341, doi:10.3390/toxins13050341; 2) Meuskens I, Saragliadis A, Leo JC and Linke D (2019) Type V Secretion Systems: An Overview of Passenger Domain Functions. Front. Microbiol. 10:1163. doi: 10.3389/fmicb.2019.0116

  1. Lines 173-176: the explanation here is a bit confusing and should be rephrased.

We have entirely rewritten the paragraph as follows: “For secretion of proteins involved in communication, virulence and adhesion, Gram-negative bacteria have evolved different secretion systems (I-IX). Some of these sys-tems are formed by multiple proteins building a complex spanning the cell envelope, whereas the type V secretion system, is rather minimal. Proteins of the Type V secretion system are known as autotransporters (ATs) to indicate a self-sufficient system for secre-tion. [28] In particular, autotransporters (ATs) are outer membrane/secreted proteins structurally characterized by the presence of three distinct domains: i) an N-terminal pas-senger domain, which typically mediates the export of the protein across the cytoplasmic membrane, ii)a surface-localized protein known as passenger or α-domain and iii) a car-boxy-terminal domain, known as β-barrel translocator domain, which facilitates the se-cretion of the passenger domain through the outer membrane. These properties allow their independent transport across the bacterial membrane system and their final routing to the cell surface. [29]”

  1. Lines 223-224: The species is in point of fact Salmonella enterica (as noted correctly on line 82). Typhimurium is a serotype of S. enterica, so the correct notation would be S. enterica serotype Typhimurium, or at the very least S. Typhimurium (the serotype is not given in italics).

Thank you for the remark. We modified the text as suggested.

  1. Lines 283-287: The explanation is again a little confusing. Are the authors suggesting that a single molecule inhibiting both be developed, or that both inhibitors should be applied at the same time? Please clarify what is meant here.

We modified the sentence, we reckon that the meaning is clearer now.

Reviewer 2 Report

Providing justifications will not help.

Please provide proper answers, and these must be clearly mentioned in response to the reviewer's letter

Author Response

Please provide proper answers, and these must be clearly mentioned in response to the reviewer's letter

Q1. Title conveys “ …fight against antibiotic resistance.”

The proposed antiadhesion strategy and the associated details do not elucidate how it will help to counter antibiotic resistance. Hence, the title needs to be revised to match the contents of the article.

Thank you for the suggestion. We modified the title in “Recent development in the inhibition of bacterial adhesion as promising anti-virulence strategy”

Q2.  Like the emergence of antibiotic Resistance, there is always a possibility of bacteria undergoing mutation and developing resistance to anti-adhesion agents.

We modified the text following the reviewer’s suggestion, by adding a brief discussion on the possibility of bacteria undergoing mutation, at the end of the introduction: “Even if it has been hypothesized that anti-virulence agents will cause much weaker selection for resistance than the traditional antibiotics, some studies demonstrated that bacteria can mutate and develop resistance to anti-adhesion agents. However, the existence of mechanisms of resistance does not necessarily mean that they will spread becoming a clinical problem.” We also added ref [10]

Q3. How many factors regulate the adhesion process? What is the frequency with which adhesion negative mutants emerge and undergo reversion.

It is very difficult to answer this question, the adhesion process is a very complicated and dynamic process.

We added a brief discussion on this topic in the conclusion:

“In order to develop an effective anti-adhesion strategy it is necessary to fully understand the mechanisms of pathogen-host interaction, which is a very complex and dynamic process. To obtain a successful therapy, it may be useful to administer combinations of different pathoblockers, capable of acting against different virulence factors, or combinations of these with traditional antibiotics to reduce the bacterial load in high-risk patients.”

Q4. Lines 56-57

Both Gram-positive and Gram-negative pathogens have a plethora of proteins and protein assemblies, anchored to their cell walls, involved in the adhesion to the host tissues.

How many proteins and protein assemblies do we need to handle to develop a viable anti-adhesion strategy?

As we reported in the manuscript, different proteins can be considered target to develop an anti-adhesion strategy. We modified the text both in the introduction and in the conclusions as follow:

Introduction: “Adhesins, such as many other virulence factors, are often limited to closely related pathogens and that entails that anti-adhesion agents showed usually a narrow spectrum. For this reason, a combination of several agents may be needed to obtain an efficacious treatment against persistent pathogenic bacteria, but, on the other hand, it was observed that the narrow spectrum of most approaches compromises horizontal gene transfer, which is considered a major process for the transmission of resistance”

Conclusion:“The inhibition of this transpeptidase, in addition to the previously mentioned advantages, has another benefit: hitting a single target results in the inhibition of numerous proteins involved in adhesion (MSCRAMMs).”

To highlight that, in the case of Gram-positive bacteria, for example, the inhibition of SrtA leads to the inhibition of numerous other proteins (MSCRAMMs) involved in the adhesion. In this way, hampering only one enzyme we can have a significant inhibition of different targets involved in the adhesion.

Q5.  Lines 60-62

Most bacterial adhesins, in both Gram-negative and Gram-positive bacteria, are organized as thin thread-like organelles called fimbriae or pili, which are involved in many important bacterial processes, including conjugation, adherence, twitching motility, biofilm formation and immunomodulation.[10]

Anti-adhesin may not affect the growth but is likely to put selective pressure on other bacterial processes.  

We modified the text (lines 54-55) as suggested: “Although anti-adhesins could put selective pressure on some bacterial processes, since adhesion is not required for microbial survival, its inhibition can be considered a promising approach to treat multidrug resistant infection…”

Q6. Lines 72-76

DsbA, was identified as an attractive strategy to attenuate the virulence of relevant pathogens, such as E. coli and Salmonella enterica. In Gram-negative pathogens, in fact, DsbA assumes a key role in the bacterial pathogenesis by catalyzing disulfide bond formation in the production of several bacterial proteins, including adhesins, flagellae, toxins, and other virulence factors [13]

The involvement of DsbA in several bacterial proteins involved in the virulence of relevant pathogens is a cause of concern since it can generate high selective pressure on bacterial life processes.

We modified the text as follows: “Despite, the involvement of DsbA in several virulence factors of relevant pathogens is a cause of concern since it can generate high selective pressure on bacterial life processes, these enzymes, being involved in the biogenesis of adhesins, currently represent interest-ing target candidates to obtain anti-virulence agents with anti-adhesion mechanism.”

Q7.  Line 86: Table 1. Examples of the main MSCRAMMs with adhesive properties.

It implies that multiple factors need to be inhibited to achieve an anti-adhesion target.

If the target is the genes inhibition or intracellular molecules then it is difficult to stop the production of adhesive molecules.

The target should be the molecules released in the vicinity of the pathogenic bacteria.

Thank you for the interesting comment. We report the SrtA inhibition as strategy to impeed the covalent anchoring of MSCRAMMs on the bacterial cell wall. We modified the text highlighting the possibility to inhibit multiple factors to obtain an effective anti-adhesion strategy.

Q8. Lines 236-237: Among the different strategies proposed in the last years, the antivirulence approach may be considered one of the most promising.

This proposal/opinion has been made by various researchers. In the last three decades, quite a few Review and Opinion articles have focused on it as one of the most promising candidates for developing “antivirulence” strategies.

So, it is recommended that this article may not be categorized as an Opinion article. Instead, it should focus on providing an update on the Anti-adhesion research status. And suggest whether it can hold well as a standalone strategy or need complementation with other processes.

We agree with the reviewer's observation. We prepared this manuscript as invited Mini-review for the special issue "Enzymes and Enzyme inhibitors-Applications in Medicine and Diagnosis 2.0". Unfortunately, despite the guest Editor had agreed to this type of contribution, the journal does not provide this type of article, therefore we had to choose an article type that could fit. For what concern the suggestion on a possible therapeutic application we added a sentence in the conclusion.

Q9. References

 Only 1 Reference pertains to those published in 2022.

And only 5 References are from 2021.

If the interest in this area is declining among scientists, then it reflects that the area is losing its importance.  

We added other five new recent references (Molecular Aspects of Medicine 2021, 81, 100998; Antibiotics 2023, 12, 195; Bioorganic Chemistry 2023, 131, 106307, World J Microbiol Biotechnol 2022, 39, 18; Toxins 2021, 13, 341)

Round 3

Reviewer 2 Report

The Review is overstating the theme.

 A simpler and smaller version emphasizing that anti-adhesion can contribute to other strategies can make a better presentation.  

Author Response

The Review is overstating the theme.

 A simpler and smaller version emphasizing that anti-adhesion can contribute to other strategies can make a better presentation

A simpler and smaller version was our original idea, in fact, the previous version of the manuscript was noticeably shorter. We reckon that our manuscript has greatly benefited from the Reviewers ‘evaluation but, inevitably, at the expense of simplicity and shortness. Anyway, we tried to semplify the manuscript being careful not to remove the modifications previously requested by the referees. We sincerely hope that it finally be considered by the reviewer worthy of publication in the journal.